# Identification of SNPs and Candidate Genes Associated with Salt Stress in Two Korean Sorghum Cultivars and Understanding Selection Pressures in the Breeding Process

**Donghyun Jeon** [1] , **Solji Lee** [2], **Sehyun Choi** [2], **Yuna Kang** [2] **and Changsoo Kim** [1,2,*]

1 Department of Science in Smart Agriculture System, Chungnam National University, Daejeon 34134, Korea
2 Department of Crop Science, Chungnam National University, Daejeon 34134, Korea
* Correspondence: changsookim@cnu.ac.kr

**Abstract:** One of the abiotic stresses, salt stress, has an impact on the production and development of crops around the world. Sorghum is a functional genomics model crop of C4 plants due to its small genome size, and it is suitable for providing a clue to the mechanism associated with salt tolerance at the transcriptomic level. However, the mechanism of salt-related genes in sorghum has not been well described. RNA sequencing, using QuantSeq, was performed on two Korean cultivars, 'Sodamchal' and 'Nampungchal', which are known to have different intensities in response to salt stress, between a control and high-salinity treatment over a different time-course. In addition, physiological responses such as the proline, anthocyanin, chlorophyll, and reducing sugar contents were evaluated under the salt-stress treatment between these two sorghum cultivars. Moreover, differentially expressed genes (DEGs) between the Nampungchal and Sodamchal cultivars were identified in their leaves and roots, respectively. Moreover, the function of DEGs was confirmed through GO classification and KEGG pathway. We also analyzed the correlation between the selection pressure with DEGs by identifying Ka/Ks of DEGs. In the breeding process, the role of positive or negative selected genes was analyzed. Therefore, a new hypothesis on selection pressure was proposed from the breeding perspective of cultivars. A comparative analysis of the two sorghum cultivars provides candidate genes involved in the salt-stress response and may offer a better understanding of the salt-tolerance mechanism in sorghum.

**Keywords:** proline; anthocyanin; chlorophyll; reducing sugar; DEGs; selection pressure

## 1. Introduction

By 2050, the world's human population will reach 9.6 billion, which demands high agricultural production [1]. Therefore, crop productivity needs to increase by 70% to meet the growing demands of the population increase. However, this is very difficult due to limited agricultural land, irregular environmental conditions, and the occurrence of abiotic stresses related to climate change [2]. Among the abiotic stresses, salt stress has a major impact on the global productivity of agriculture. It is estimated that about 6.0% of the world's land, more than 800 million hectares, suffers from salt damage or soil salinization [3]. Salinity accumulation is primarily associated with inadequate drainage and saline groundwater and irrigation water based on a dry climate. The impact of salinity on crop productivity is more severe in arid and semi-arid areas where limited precipitation, a lot of evapotranspiration, high temperatures, bad water quality, and awful soil-management practices are observed [4].

Soil salinity has a negative effect on plant function, development, and yield. According to previous studies, exposure to salinity in sorghum resulted in a decrease in live above-ground biomass, the root biomass, the shoot-to-root ratio, the height, and the percentage of live leaves. In addition to phenotypic changes, the salinity-stress effects of sorghum include various molecular and physiological responses, such as changes in chloroplast

content in foliar surfaces and $Na^+$ and $K^+$ absorption [5]. The reason for such a response is that salinity increases osmotic stress. Accordingly, leaf appearance and growth are reduced, stomata are closed, and photosynthesis is reduced. Moreover, as $Na^+$ accumulates in the plant, the absorption of $K^+$, which is an essential ion for maintaining biological functions, decreases [3]. Salinity reduces plant growth and development because of excessive uptake of sodium ($Na^+$), chloride ($Cl^-$), and nutritional imbalance. Salt stress causes changes in various physiological and metabolic pathways, depending on the intensity of the stress. Salinity stress generally induces oxidative stress due to the production of reactive oxygen species (ROS) such as singlet oxygens, superoxides, hydroxyl radicals, and hydrogen peroxide, disturbing the essential cellular functions of plants [6]. In the early stages of salt stress, which accumulates high salt in the soil, plants reduce the water-absorption capacity of the root system and accelerate water loss from the leaves. Thus, salt stress is also regarded as hyperosmotic stress [7]. Osmotic stress in the early phase of salinity stress causes various physiological changes, such as the destruction of the membrane, a nutrient imbalance, and a reduction in the detoxifying ability for ROS. Moreover, osmotic stress reduces the antioxidant enzymes and decreases the photosynthetic activity and pore-opening capacity [8]. When both $Na^+$ and $Cl^-$ enter the cells, serious ionic imbalances occur, and excessive absorption can cause physiological damage. A high $Na^+$ concentration reduces the productivity and the absorption of $K^+$ ions, which are essential for growth and development.

Plants have developed various mechanisms to maintain productivity in salt-stressed environments [9]. The salt-stress response of plants is regulated by multiple genes that contribute to the tolerance [10]. Salt stress also induces a wide range of changes in gene expression in morphological, physiological, biochemical, cellular, and molecular pathways. These mechanisms are controlled by various transcription factors and genes such as ion transport, senescence-associated response, compatible solute synthesis, the antioxidant system, hormonal regulation, $Ca^{2+}$ signaling, and SOS signaling pathways [11]. As the availability of sequence data has increased, attempts to identify genes associated with the salt-stress response have been conducted in recent years in many plant species. A common strategy for identifying genes associated with salt stress involves using a comparative study of various genotypes of tolerance to abiotic stresses. Comparative studies between salt-sensitive and salt-tolerant genotypes have been conducted for Arabidopsis [12], rice [13], olive [14], populous [15], and tomato [16] plants.

Sorghum (*Sorghum bicolor* (L.) Moench) is an annual crop that originates from Africa and is now cultivated in tropical and subtropical regions around the world [17]. Sorghum is the world's fifth-most cultivated cereal crop after rice, wheat, maize, and barley. It remains an essential element of food security for more than 300 million people in Africa [12]. One of the most crucial factors of sorghum productivity is C4 photosynthesis, which involves biochemical and morphological specialties that increase carbon assimilation at high temperatures [14]. Sorghum accumulates biomass efficiently and grows faster than other grain crops. It is also considered one of the most promising bioenergy crops due to the abundant fermentable sugar in its stalk [18]. In addition, its small genome size of about 730 Mb makes sorghum an attractive model of Saccharinae and other C4 grasses for functional genomics [19]. However, only a few candidates have been identified as salt-responsive genes, although sorghum is a tolerant crop to salt stress.

In this study, we investigated the responses of two Korean sorghum cultivars under salt stress, using QuantSeq. This study identified the difference between the two cultivars at a high concentration of salt (150 mM NaCl) compared to the non-treated samples from the leaf and root tissues at the seeding stage. The aims of this study were (I) to examine the difference in physiological responses of the salt-tolerant and susceptible cultivars, (II) to identify genes that are expressed differentially between the two cultivars, and (III) to analyze mutations that cause functional changes in the gene sequences. Finally, by linking these results, the mechanism for salt stress was comprehensively analyzed. The results of

this study will help identify the overall response and mechanism of salt stress in sorghum and may serve as the basis for future breeding and genomics research of sorghum.

## 2. Materials and Methods

### 2.1. Plant Materials and Stress Treatment

Two sorghum cultivars, Nampungchal and Sodamchal, were used as plant materials in this study. It was reported that Nampungchal showed superior growth in reclaimed lands compared to Sodamchal [20]. Nampungchal was made through pure line selection from a native species of 'Namhae' in Gyeongsangnamdo, the Republic of Korea, and has features such as a lodging tolerance, multi-environmental adaptability, and high yielding ability. On the other hand, Sodamchal was bred by crossing 'Hwangguemchal' and 'Jungmo 4001'. As a short-statured cultivar, Sodamchal has a lodging tolerance and is advantageous for mechanized harvesting. Seeds were obtained from the Rural Development Administration (Jeonju, Republic of Korea). Before the test, all seeds were sterilized by a thiram–benomyl mixture (5 g/L) for 24 h. Plant growth and stress treatment were conducted in a greenhouse at Chungnam National University (CNU), Republic of Korea (36°22′06.6″ N 127°21′11.3″ E) from June to July 2019. Seeds of both cultivars were grown in plastic pots (6 cm deep, with a 1.5 cm diameter) filled with bed soil until the seeding stage and then transferred to new, wider pots (8 cm deep with a 7 cm diameter) filled with sterilized vermiculite. A half-strength Hoagland nutrient solution was provided through a reservoir under the pots, and the plants absorbed nutrients through capillary action. Plants were grown to adapt for seven days in a normal half-strength Hoagland nutrient solution. After that, the plants were selected uniformly by the size of the plant before the stress treatment. The stress treatment was performed with 150 mM sodium chloride in the half-strength Hoagland nutrient solution. After zero, three, and nine days of stress treatment, the plant leaves and roots were harvested for further analyses, with three biological replications.

### 2.2. Quantification of the Anthocyanin Contents

The leaves and roots of sorghum were ground into powder, using a mortar and pestle. They were mixed with five volumes of extraction buffer (45% methanol and 5% acetic acid). The mixture was allowed to spin down for five minutes at $12{,}000 \times g$ (room temperature), and the supernatant was transferred to a new tube. The absorbance was measured at 530 and 657 nm with the nanoMD UV–Vis Bio Spectrophotometer (Scinco, Republic of Korea), and the anthocyanin contents were calculated as follows [21]:

$$[Abs_{530}/g \text{ (fresh weight)}] = [Abs_{530} - (0.25 \times Abs_{657})] \times 5$$

### 2.3. Quantification of the Proline Contents

First, 50 mg of fresh-weight sorghum sample was ground and mixed with 1 mL of extraction buffer (40% EtOH). The mixture was left overnight at 4 °C and centrifuged at $14{,}000 \times g$ to separate the supernatant, which was transferred to a new tube. Then 1 mL of reaction agent (ninhydrin 1%:60% acetic acid:20% ethanol) was mixed with 500 uL of ethanol extract and heated at 95 °C for 20 min. After that, the absorbance was measured at 520 nm. The absorbance was quantified by using a standard curve.

### 2.4. Quantification of the Reducing Sugar Contents

The total reducing sugar was estimated by the DNS method, measuring the absorbance at 575 nm, using a spectrophotometer. DNS reagent was prepared with potassium tartrate 150 g, NaOH 8 g, and 3,5-dinitrosalicylic acid (DNS) at a final volume of 1000 mL, using ddH$_2$O. Then 300 mg of ground sorghum sample was mixed with diluted DNS reagent (2 mL DNS + 7 mL distilled water). The mixture was heated at 100 °C for 5 min, using a water bath, and then cooled down. The absorbance was measured at 575 nm and quantified through a standard curve.

### 2.5. Quantification of the Chlorophyll Contents

First, 300 mg of sorghum seedling powder was mixed with 5 mL of 80% acetone. The mixture was left in the dark for 30 min. The absorbance was measured at 663 and 645 nm. The total chlorophyll contents were calculated by using the following equation:

$$\text{Chlorophyll [a + b] (mg/g)} = [8.02 \times \text{Abs}_{663} + 20.20 \times \text{Abs}_{645}] \times \text{extract volume}/1000 \times \text{fresh weight.}$$

### 2.6. RNA Isolation

Total RNA was isolated from the leaves and roots of the two sorghum cultivars sampled at zero and three days after treatment, using Trizol reagent (Invitrogen, USA). The RNA quality was measured by the Agilent 2100 bioanalyzer, using the RNA 6000 Nano Chip (Agilent Technologies, Amstelveen, Netherlands), and RNA quantification was processed with the ND-2000 Spectrophotometer (Thermo Inc., Waltham, DE, USA).

### 2.7. Construction of the QuantSeq Library for DEG Analyses

The sequencing library was constructed by using the QuantSeq 3′ mRNA-Seq Library Prep Kit (Lexogen, Inc., Vienna, Austria) according to the manufacturer's instructions. In short, the total RNA was prepared with 500 ng, an oligo-dT primer containing an Illumina compatible sequence at its 5′ end was annealed to the RNA, and reverse transcription was conducted. Synthesis of the second strand was initiated by a random primer with an Illumina-compatible linker sequence at its 5′ end after degradation of the RNA template. The double-stranded library was purified with magnetic beads to remove all reaction components. To add the complete adapter sequences required for cluster generation, the library was amplified. The final library was purified from the PCR components. High-throughput sequencing was performed as single-end 75 sequencing, using NextSeq 500 (Illumina, Inc., San Diego, CA, USA).

### 2.8. Construction of the RNASeq Library and Sequencing for SNP Analyses

Using the NEBNext Ultra II Directional RNA-Seq Kit (NEW ENGLAND BioLabs, Inc., Hitchin, UK), sequencing libraries were prepared from the total RNA. The rRNA was eliminated by using the RIBO COP rRNA depletion kit (LEXOGEN, Inc., Vienna, Austria). The purified RNAs were used for cDNA synthesis and shearing. Indexing was carried out by using the Illumina indexes 1–12. The enrichment step was performed by using PCR. After that, the libraries were tested by using the Agilent 2100 bioanalyzer (DNA High Sensitivity Kit) to evaluate the mean fragment size. Using a StepOne Real-Time PCR System (Life Technologies, Inc., Carlsbad, CA, USA), quantification was performed by using the library quantification kit. High-throughput sequencing was performed as paired-end 100 sequencing, using Hiseq X10 (Illumina, Inc., USA).

### 2.9. Sequencing Data Analysis

Sequencing reads were aligned by using bowtie2 [22]. The reference genome used was the NCBI *Sorghum bicolor* BTx623 v3.0. The alignment files were used to assemble transcripts, predict expression levels, and find DEGs. DEGs were determined by counts, using Bedtools [23]. The read-count data were processed according to the quantile normalization method, using EdgeR within R [24], using Bioconductor [25]. Gene classification was conducted by the DAVID and Medline databases.

### 2.10. Gene Ontology (GO) and Kyoto Encyclopedia of Genes and Genomes Database (KEGG) Pathway Enrichment Analysis of DEGs

GO functional enrichment analysis was performed to identify DEGs significantly enriched in the GO terms. DEGs were subjected to singular enrichment analysis (SEA) through PlantRegMap (http://plantregmap.gao-lab.org/) [26]. Significant terms were selected with a threshold *p*-value of $\leq 0.05$. GO annotations were classified based on the biological process, molecular function, and cellular component. KEGG analysis was performed to determine the potential biological pathways of the sorghum transcriptome [27].

KEGG pathways were assigned to the genes, using the online KEGG web server (http://www.genome.jp/kegg/, accessed on 1 March 2022).

### 2.11. Analysis of Quantitative Real-Time PCR (qRT-PCR)

Total RNA was extracted from the leaves and roots of Nampungchal and treated with salt stress in three biological replicates. A total of six genes, LOC8072968, LOC8065132, LOC8077222, LOC8077861, LOC8083797, and LOC8077913, were subjected to qRT-PCR analysis. The qRT-PCR primers were prepared through Primer3 (v. 0.4.0) based on the NCBI gene sequence with a primer size of 20–25 bp, GC content of 45–55%, melting temperature (Tm) of 60–62 °C, and product size of 150 bp or less. *PP2A* was used as the reference gene [28]. Primer information is available in Supplementary File S1. cDNA synthesis was conducted by using a Compact cDNA Synthesis kit (Smart Gene, Daejeon, Korea). Then qRT-PCR was performed with the CFX Connect™ Real-Time PCR Detection System (BIORAD, Herculs, CA, USA) and the SYBR green Q-PCR Master mix (Smart Gene, Korea). PCR reactions were programmed as a 3-step cycle, template denaturation and enzyme activation at 95 °C for 10 min, followed by 40 cycles of denaturation at 95 °C for 15 s, annealing at 60 °C for 30 s, and extension 72 °C for 30 s. The resulting data were normalized to the Ct value of the housekeeping gene, and the normalized expression value was calculated with the delta–delta Ct (DDCT) method to calculate the log2 fold-change value.

### 2.12. SNP Classification and Selection Pressure

Paired-end sequencing reads obtained from the total RNAseq of Sodamchal and Nampungchal were aligned to the BTx623 reference genome, using bowtie2. Samtools was used to convert SAM to the BAM file format, and Bcftools was used to create a variant calling file (VCF) [29]. SnpEff was used to annotate the effects of the SNPs [30]. For those identified SNPs, non-synonymous substitution per non-synonymous site ($K_a$) and synonymous substitution per synonymous site ($K_s$) were evaluated. In this way, the $K_a/K_s$ ratio was calculated through the genes based on the position of the SNPs, using our in-house python script. Gene selective pressure was divided into diversifying groups ($K_a/K_s > 1$) and purifying groups ($K_a/K_s < 1$) [31]. For genes with selection pressure, the impact of the variants was confirmed through SnpEff annotation information. Furthermore, selection-pressure information was used to select candidate genes based on the list of DEGs (Figure 1).

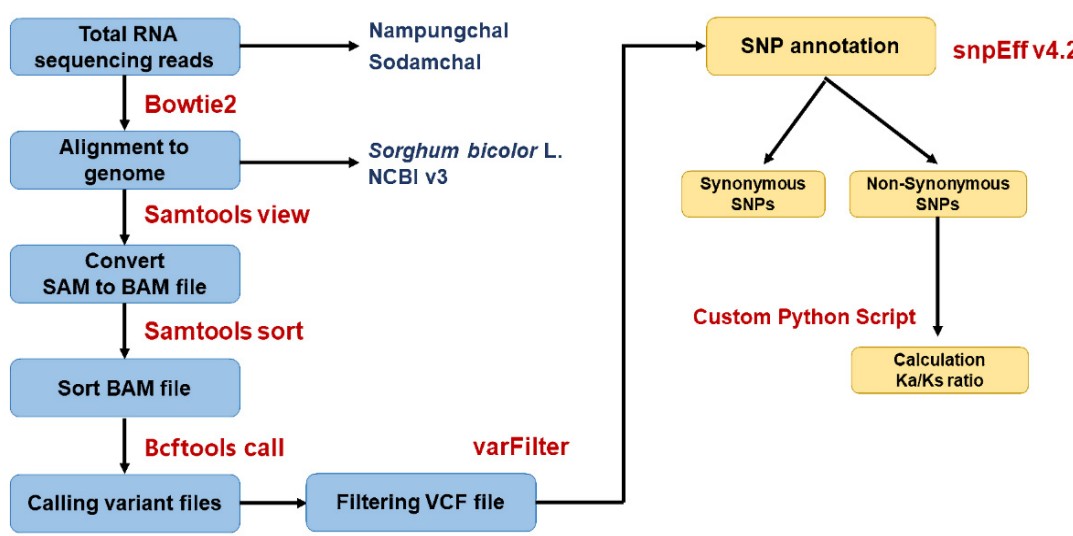

**Figure 1.** Calling and analysis workflow of SNPs, using various bioinformatics tools.

### 2.13. Statistical Data Analysis

Data are shown as the mean of three biological replications with respective standard error of the mean (SEM) bars. Differences among the means of all samples were tested with one-way ANOVA, followed by Tukey's multiple comparison test ($p < 0.05$ was considered significant), using the IBM SPSS Statistics version 24 software (IBM SPSS, Inc., Armonk, NY, USA).

## 3. Results

### 3.1. Growth Performance and Physiological Response to Salt Stress

It was reported that Nampungchal and Sodamchal have tolerance and sensitivity to salt stress, respectively, based on a previous study [20]. The two sorghum cultivars were treated with 150 mM NaCl, and those cultivars showed a clear difference in growth under salt-stress conditions, as was consistent with the previous report (Figure 2). To see the physiological responses corresponding to the growth patterns, we evaluated the contents of anthocyanin, proline, chlorophyll, and reducing sugar from the samples harvested at zero, three, and nine days after treatment (DAT).

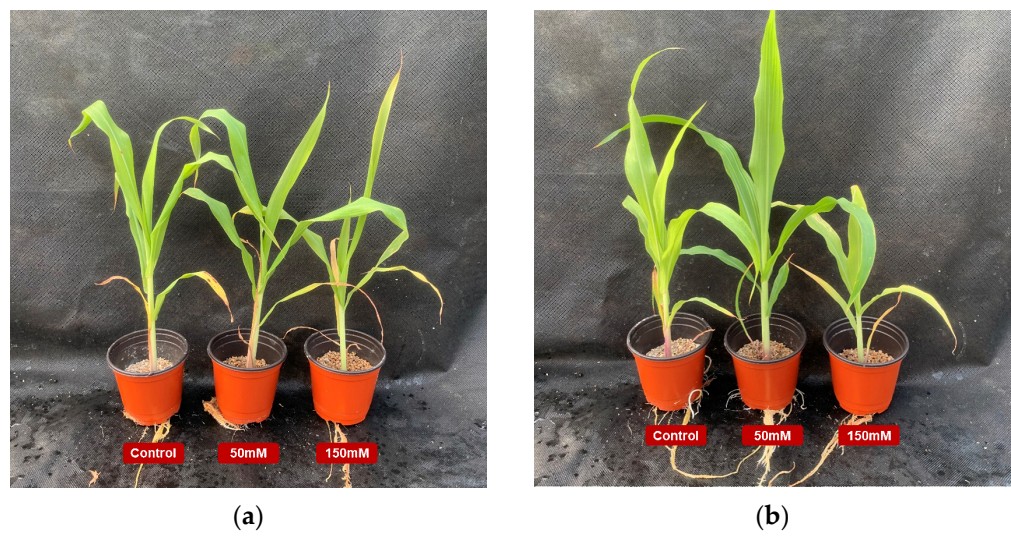

(**a**)　　　　　　　　　　　　　　　　　　(**b**)

**Figure 2.** Growth performance nine days after the salt treatment of two sorghum cultivars: (**a**) Nampungchal (salt-tolerant) and (**b**) Sodamchal (salt-sensitive).

### 3.1.1. Quantification of the Anthocyanin Contents

The total anthocyanin contents in the leaves of the Nampungchal were $0.43528 \pm 0.02237$, $0.73001 \pm 0.23309$, and $1.48755 \pm 0.54752$ on zero, three, and nine days after 150 mM salt-stress treatment, respectively. The differences among the means of each day after treatment were tested with one-way ANOVA, followed by Tukey's multiple comparison test ($p < 0.05$ was considered significant). The anthocyanin contents were significantly increased after nine days of salt-stress treatment compared to before the salt-stress treatment in Nampungchal. On the other hand, in the leaves of Sodamchal, the total anthocyanin contents were $0.32214 \pm 0.02535$, $0.73237 \pm 0.0518$, and $0.40554 \pm 0.04154$ at zero, three, and nine days after the salt-stress treatment, respectively. The total anthocyanin contents increased three days after the salt-stress treatment, which is the initial phase of the salt stress. However, there was no significant difference in the anthocyanin contents between the before stress treatment and after nine days of salt-stress treatment in Sodamchal (Figure 3A). The total anthocyanin contents of Nampungchal in the roots were $0.10989 \pm 0.00891$, $0.16779 \pm 0.02537$, and $0.47956 \pm 0.13209$ at zero, three, and nine days after the salt-stress treatment, respectively, while in the roots of Sodamchal, they were $0.0926367 \pm 0.02115$, $0.1202033 \pm 0.04545$, $0.3896767 \pm 0.11921$ at zero, three, and nine days after the salt-stress treatment, respectively (Figure 3B). In the roots, the anthocyanin con-

tents of both Nampungchal and Sodamchal were increased significantly nine days after the salt-stress treatment. However, there was no significant difference between the two cultivars on the ninth day.

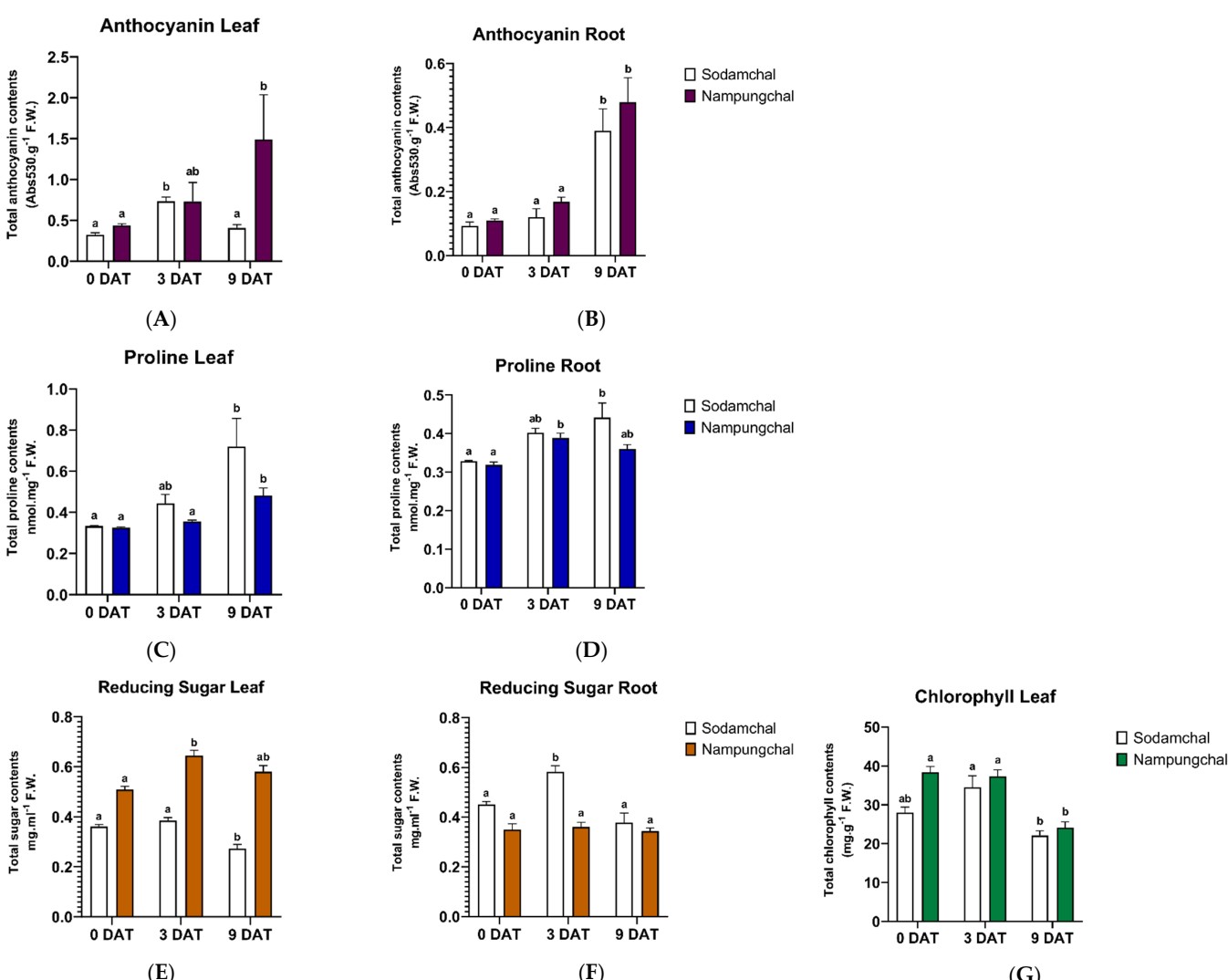

**Figure 3.** Quantification of physiological and biological responses of the two sorghum cultivars, Nampungchal and Sodamchal, under salt stress (DAT, days after treatment): (**A**) anthocyanin in the leaves, (**B**) anthocyanin in the roots, (**C**) proline in the leaves, (**D**) proline in the roots, (**E**) reducing sugar in the leaves, (**F**) reducing sugar in the roots, (**G**) and chlorophyll in the leaves. The *y*-axis represents the total contents of the physiological and biological responses, and the *x*-axis represents the time after NaCl treatment (days after treatment). Values represent the mean $\pm$ SEM (*n* = 3). The different letters indicate significant differences among different days, using Tukey's test at *p* < 0.05.

### 3.1.2. Quantification of the Proline Contents

The measurements of the proline contents were made in the leaves and roots at zero, three, and nine days after the salt-stress treatment. The amount of proline in the leaves of Nampungchal was 0.32598 $\pm$ 0.00503, 0.35519 $\pm$ 0.12185, and 0.4824 $\pm$ 0.06269 at zero, three, and nine days, respectively, and the proline contents were significantly increased nine days after the salt-stress treatment. The proline contents in the leaves of Sodamchal were 0.33335 $\pm$ 0.00555, 0.44406 $\pm$ 0.07459, and 0.72025 $\pm$ 0.2362 at zero, three, and nine days after the salt-stress treatment, respectively. The proline contents of Sodamchal in the leaves were also significantly increased nine days after the salt-stress treatment (Figure 3C). The proline contents were also measured in the roots. Under salt stress, the proline

contents of Nampungchal in the roots were 0.31872 ± 0.01281, 0.38936 ± 0.02215, and 0.36018 ± 0.01891 at zero, three, and nine days after the salt-stress treatment, respectively. After three days of the salt-stress treatment, the proline contents of Nampungchal in the roots were increased, but after nine days, the proline contents did not differ from those of the non-treatment (zero days after treatment). The proline contents in the Sodamchal roots were 0.32832 ± 0.00354, 0.40229 ± 0.01984, and 0.40229 ± 0.01984 at zero, three, and nine days after the salt-stress treatment, respectively. The proline contents of the Sodamchal roots were significantly increased nine days after the salt-stress treatment, showing different patterns from Nampungchal.

### 3.1.3. Quantification of the Reducing Sugar Contents

The reducing sugar contents of the two sorghum cultivars were measured in both the leaves and roots. The reducing sugar contents were measured at zero, three, and nine days after the salt-stress treatment of Nampungchal in the leaves. The values were 0.50832 ± 0.02285, 0.64405 ± 0.03768, and 0.56829 ± 0.04212, respectively. The reducing sugar contents of the Nampungchal leaves tended to increase in the early stage of the salt stress (third day) and were maintained at a similar level until nine days after the treatment. On the other hand, the reducing sugar contents in the Sodamchal leaves were 0.36066 ± 0.01418, 0.3841 ± 0.0217, and 0.27283 ± 0.029 at zero, three, and nine days after the salt-stress treatment, respectively (Figure 3E). In the Sodamchal leaves, unlike in the Nampungchal leaves, the reducing sugar contents were significantly decreased nine days after the treatment. The reducing sugar contents of Nampungchal in the roots were 0.34917 ± 0.04104, 0.36001 ± 0.03306, and 0.34344 ± 0.02191 at zero, three, and nine days after the salt-stress treatment, respectively. Each of these values did not show a significant difference among the sampling dates. On the other hand, in the roots of the Sodamchal, the reducing sugar contents were 0.45129 ± 0.01939, 0.58293 ± 0.04162, and 0.37774 ± 0.06661 at zero, three, and nine days after the salt-stress treatment, respectively. The reducing sugar contents in the roots of Sodamchal tended to significantly increase in the initial stage after receiving salt stress (third day). However, as the stress continued, it decreased, and there was no significant difference compared to before the stress (Figure 3F).

### 3.1.4. Quantification of the Chlorophyll Contents

The chlorophyll contents in Nampungchal were 38.35589 ± 2.6084, 37.28562 ± 2.95804, and 24.10424 ± 2.6247 at zero, three, and nine days after the salt-stress treatment, respectively. The chlorophyll contents in Sodamchal were 28.0130606 ± 2.42866, 34.4684008 ± 5.16737, and 22.10197340 ± 2.0821 at zero, three and nine days after the salt-stress treatment, respectively. In both cultivars, the chlorophyll content decreased over time (ninth day) compared to the initial salt stress (third day). There was no significant difference in the chlorophyll contents between the two cultivars on the ninth day (Figure 3G).

### 3.2. *Analysis of Gene Expression in Two Sorghum Cultivars Using QuantSeq Data*

We obtained information on gene expression under the salt treatment in both leaf and root tissues using two sorghum cultivars. DEGs were identified between the two sorghum cultivars and the direction of expression of a specific gene was identified by qRT-PCR. Also, gene description was identified through GO ontology and KEGG pathway analysis.

### 3.2.1. Differences in Gene Expression Profiles of the Two Sorghum Cultivars under Salt Stress

Differences in gene expression with or without salt stress can help us understand the stress-responsive mechanisms of the two sorghum cultivars. We identified DEGs by comparing the transcriptomes under salt-stress conditions. The reads generated by QuantSeq were aligned to the BTx623 sorghum reference genome. DEGs were identified through four comparative combinations, with a total of eight samples. The four DEGs' analysis combinations were the stress-treated Nampungchal and Sodamchal leaves (NLs/SLs),

the stress-treated Nampungchal and Sodamchal roots (NRs/SRs), the non-stressed Nampungchal and Sodamchal leaves (NLc/SLc), and the non-stressed Nampungchal and Sodamchal roots (NRc/SRc). A total of 5903 DEGs were identified in the four combinations. The highest number of DEGs (1804) were identified between the roots of the Nampungchal and Sodamchal cultivars under salt-stress conditions (NRs/SRs). Moreover, the lowest number of DEGs (1186) was in the roots of the Nampungchal and Sodamchal cultivars under control conditions (NRc/SRc). In addition, the number of DEGs was 1270 in the leaves of the control (NLc/SLc), and the number of DEGs was 1643 in the salt-stress conditions of the leaves between the Nampungchal and Sodamchal cultivars (NLs/SLs). In both the leaves and roots, the number of DEGs is higher in the stress condition than in the control. In the stress conditions, downregulated genes were considerably higher compared to the upregulated genes in both the leaves and roots between the two sorghum cultivars (NLs/SLs and NRs/SRs). For a better understanding, we confirmed the distribution of the DEGs with a Venn diagram. As a result, 148 DEGs were identified as common DEGs in all the comparisons. Among them, 58 were upregulated genes, and 88 were downregulated genes, and two were contra-regulated genes which are expressed contrastively in each group (Figure 4). To identify candidate genes related to salt tolerance, we identified genes commonly upregulated in the leaves and roots between the Nampungchal and Sodamchal cultivars under salt stress. There are a total of 152 genes that are commonly upregulated in the leaves and roots.

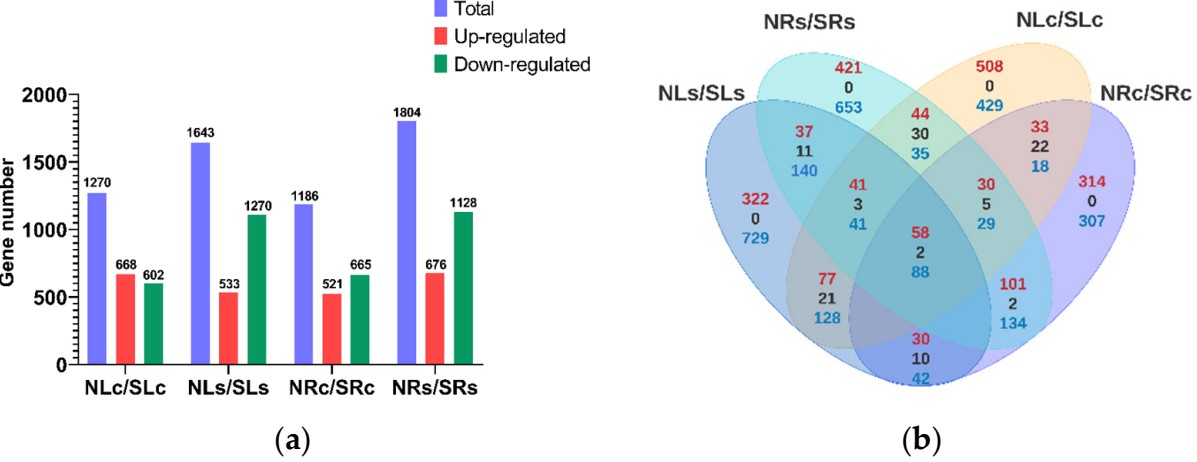

(**a**)                                                  (**b**)

**Figure 4.** Number of DEGs: (**a**) DEGs in the four comparative combinations of Nampungchal and Sodamchal in the leaves and roots under salt-stress and control conditions and (**b**) distribution of overlapping or unique genes identified by Venn diagram. NL(R)c, Nampungchal leaves (roots) control; SL(R)c, Sodamchal leaves (roots) control; NL(R)s, Nampungchal leaves' (roots') salt-stress; SL(R)s, Sodamchal leaves' (roots') salt-stress. Number in red is number of upregulated genes, number in black is number of contra-regulated genes, and number in blue letter is number of downregulated genes.

### 3.2.2. GO Classification and KEGG Pathway Analysis of the DEGs between the Two Sorghum Cultivars

GO and KEGG pathway analyses were performed by using 152 DEGs that were simultaneously upregulated in both tissues of Nampungchal in contrast to Sodamchal under salt conditions. GO analyses were conducted through PlantRegMap, a web-based database and tool, to evaluate the specific functional categories of the genes. Among 152 genes, 75 genes were linked to GO annotations and enriched in 23 terms (*p*-value $\leq 0.05$). The sets of genes directly or indirectly related to the salt-stress response were classified as biological process, cellular component, and molecular function (Figure 5a). Among them, 14 terms were enriched in GO biological process, and 8 terms were enriched in GO molecular function. The most frequently linked term in GO molecular function was 'GO:0017111, a nucleoside-

triphosphatase activity', linked with five genes. Moreover, only one term, 'GO:0048500, a signal recognition particle', was identified in the GO cellular component, with two genes. KEGG pathway analysis was conducted to identify the biological pathways of the DEGs (Figure 5b). DEGs were linked to a total of 21 upper pathways, and the pathways with the most genes were the 'molecular pathways', which consisted of the following six sub-pathways: 'HD domain-containing protein 2 homolog', 'hydroxycinnamoyltransferase 4', 'chlorophyll ab binding protein 1', 'chloroplastic, chlorophyll ab binding protein 2, chloroplastic', 'bifunctional fucokinase/fucose pyrophosphorylase', and 'putative amidase C869.01'. Through an analysis of the GO and KEGG pathways, comprehensive molecular functions and pathways of the candidate genes were identified.

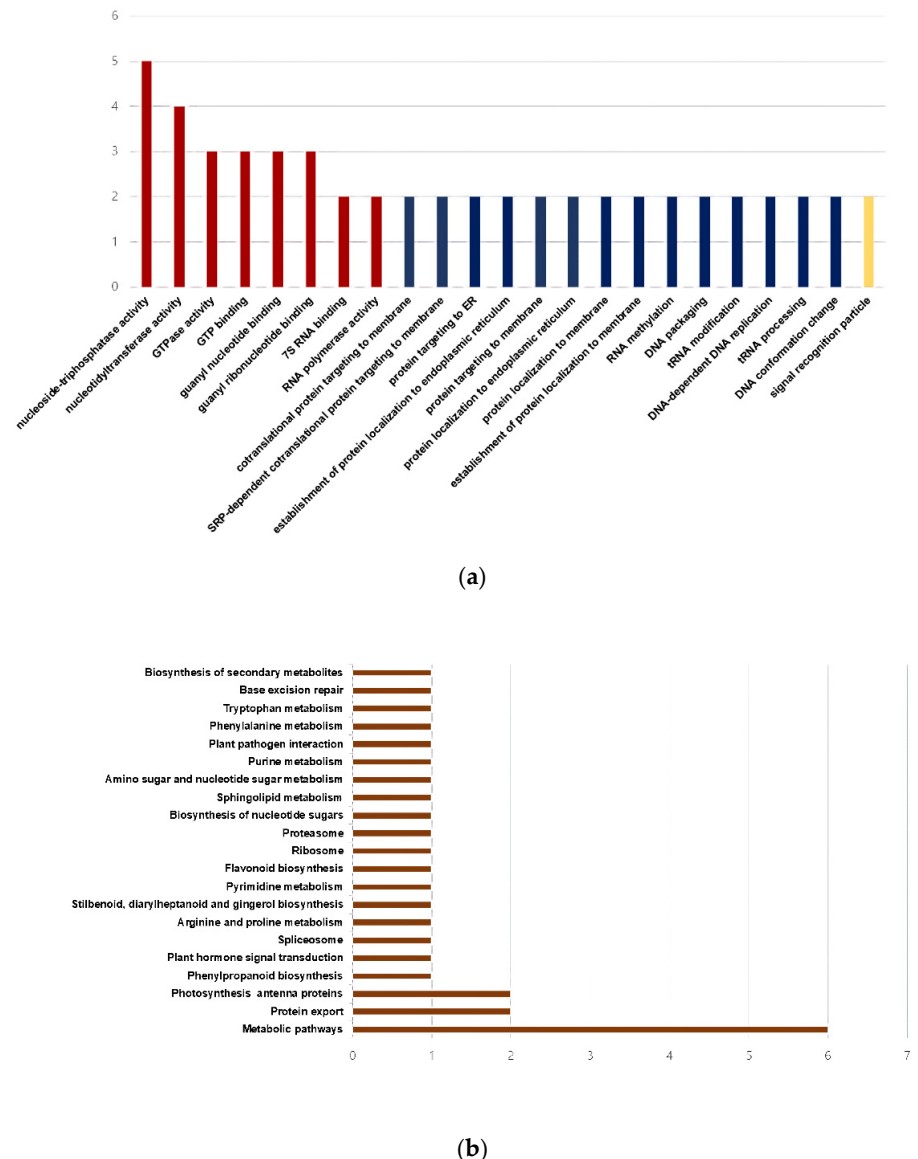

(**a**)

(**b**)

**Figure 5.** Functional annotation of simultaneously upregulated DEGs in the leaves and roots under salt stress between the two sorghum cultivars. (**a**) Gene annotation by GO analysis of DEGs. Gene annotation categories were composed of biological processes (red bars), molecular functions (blue bars), and cellular components (yellow bar). The *y*-axis represents the number of GO terms associated with the DEGs. The *x*-axis represents the GO terms. (**b**) Functional classification and pathway assignment of the identified DEGs by KEGG pathway. The *y*-axis represents the KEGG pathway, and the *x*-axis the number of KEGG pathways associated with the DEGs.

### 3.2.3. Validation of Differentially Expressed Genes by qRT-PCR

To validate the results of QuantSeq, DEGs were randomly selected to perform qRT-PCR analysis. A total of six DEGs, 'glucan endo-1,3-beta-glucosidase 14' (LOC8072968), 'xyloglucan endotransglucosylase/hydrolase protein 22' (LOC8065132), 'chlorophyll a-b binding protein 2' (LOC8077222), 'probable isoprenylcysteine alpha-carbonyl methylesterase ICME' (LOC8083797), and 'dehydrin DHN1' (LOC8077913), were selected. Of these, four genes were upregulated, while one gene was downregulated, and the qRT-PCR results showed the same expression pattern as the QuantSeq results. The genes LOC8065132, LOC8072968, LOC8077861, LOC8083797, and LOC8077913 were consistently upregulated in the qRT-PCR and QuantSeq results. Moreover, LOC8077222 was likewise consistently downregulated in the qRT-PCR and QuantSeq analyses (Figure 6).

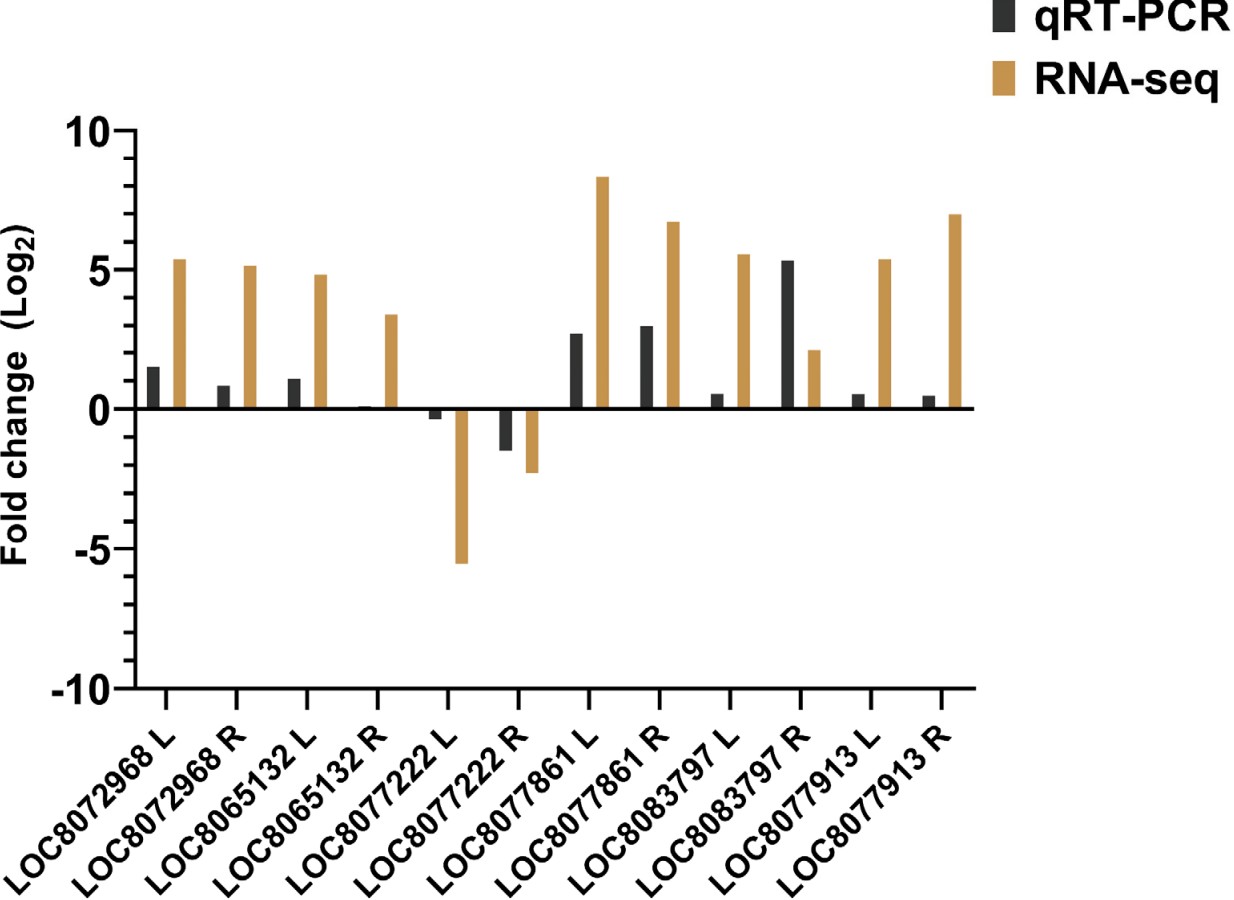

**Figure 6.** Comparison of the expression patterns of the six genes between the qRT-PCR and RNAseq results. For fold change, gene expressions between the two sorghum genotypes were obtained from the leaves and roots, respectively, under salt stress. The *y*-axis represents the relative expression level of a gene in fold change (Log$_2$). The *x*-axis represents the gene accession. L, leaf; R, root. Fold change (Log$_2$) represents the expression level of Nampungchal compared to Sodamchal, which was upregulated when it was greater than 0 and downregulated when it was less than 0. If the expression direction (up or down) of the two data is the same, it indicates that there is a correlation with each other.

### 3.3. Variants Analysis in Two Sorghum Cultivars

We obtained variants information from the total RNAseq data of leaves of two sorghum cultivars compared to the reference, respectively. The acquired variants information

was classified according to location and function, and information on selection pressure was identified.

### 3.3.1. Identification and Classification of SNPs from the Two Sorghum Cultivars

Total RNAseq reads of Nampungchal and Sodamchal were aligned to the BTx623 reference genome, and variants were generated by Bcftools. Variants were classified according to their function, location, and effect by SnpEff. Thus, 37,388 and 29,938 SNPs were obtained from the two cultivars of sorghum, respectively. The SNPs of Nampungchal approximately consisted of 11,633 non-synonymous SNPs and 13,703 synonymous SNPs. On the other hand, the SNPs of Sodamchal consisted of 8508 non-synonymous SNPs and 9607 synonymous SNPs (Table 1). We classified the variants into four categories according to their overall impact. The four categories were high, moderate, modifier, and low (Figure 7). The high-impact variants directly affect the function of a gene and are generated by a stop codon gain or loss. Stop codon gain and loss will probably result in a high level of protein functional changes due to the point mutation of the transcript. The low-impact variants mainly consist of synonymous SNPs, which cause a change in base sequence but do not change amino acids. Therefore, they have a low influence on protein expression. The moderate-impact variants consist of 'missense variants', which impact protein function through amino acid changes. Variants with a modifier impact affect gene function and consist of two untranslated regions (3′ and 5′ UTRs) and intergenic variants [32]. The impact of the variants in Nampungchal is as follows: modifier 87%, low 6%, moderate 5%, and high 2%. On the other hand, Sodamchal has some differences as follows: modifier 87%, low 7%, moderate 4%, and high 2%. Although similar in composition, the salt-tolerant Nampungchal has more high-, moderate-, and modifier-impact variants that affect gene function because the total number of variants is higher than that of the salt-sensitive Sodamchal. Among the variants of Nampungchal, the 'splice donor variant & intron variant' ranks the highest with 1554 in the high category. In the moderate-impact category, the highest term of variants is the 'Missense variant', at 7778 SNPs; in the modifier category, the highest term is the '3 prime UTR variant', at 7907; and finally, in the low category, the highest term is the 'Synonymous variant', at 9700 (Figure 8a). Similarly, in Sodamchal, the largest number of terms in each impact category is the same as Nampungchal. The number of 'Splice donor variant & intron variant' is 1663, which is the largest number of terms in the high category in Sodamchal. The highest term of variants in the moderate category is the 'Missense variant', at 5864. The highest term in the modifier category is the '3 prime UTR variant', at 5687. Finally, the highest term in the low category is the 'Synonymous variant', at 6949 (Figure 8b). When comparing the SNPs of the two sorghum cultivars, the number of 'Missense variants' affecting protein-expression changes is higher in the salt-tolerant cultivar Nampungchal than in the salt-sensitive cultivar Sodamchal.

**Table 1.** Distribution of variants in two sorghum cultivars based on gene functional motifs.

| Type of Genotypes | Type of Region | Type of Effect | Frequency of Variants |
|---|---|---|---|
| Nampungchal | Coding | Synonymous | 13,968 |
| | | Non-synonymous | 11,527 |
| | UTR | | 16,732 |
| | Splice region | | 4624 |
| | Intergenic | | 2771 |
| | Intron | | 25,534 |
| Sodamchal | Coding | Synonymous | 9600 |
| | | Non-synonymous | 8431 |
| | UTR | | 12,215 |
| | Splice region | | 10,239 |
| | Intergenic | | 2276 |
| | Intron | | 17,008 |

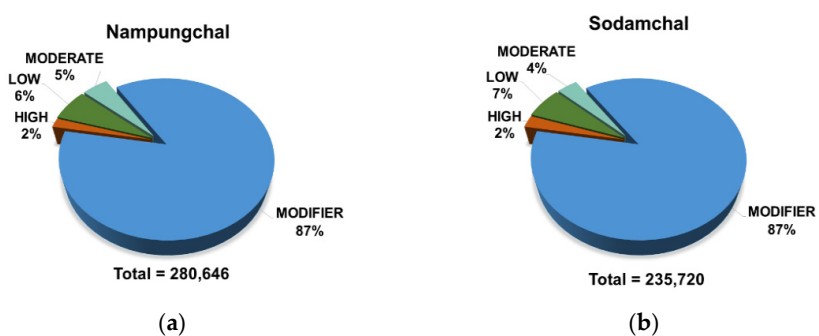

**Figure 7.** Distribution of the variants by impact consisting of high, moderate, modifier, and low: (**a**) Nampungchal and (**b**) Sodamchal.

**(a)**

**(b)**

**Figure 8.** Distribution of the subcategories of the variants classified by impact. The *y*-axis represents the subcategories of the variants, and the *x*-axis represents the number of the variants corresponding to the subcategories: (**a**) Nampungchal variants in the high, moderate, modifier, and low category; (**b**) Sodamchal variants in the high, moderate, modifier, and low category. NP, Nampungchal; SD, Sodamchal.

### 3.3.2. Identification of DEGs with Non-Synonymous SNPs

Among the variants, non-synonymous SNPs directly affect gene function by modifying the structures of protein. By comparing non-synonymous SNPs with DEGs, it is possible to narrow down the range of specific genes that increase salt tolerance in salt-tolerant cultivars. We identified DEGs carrying non-synonymous SNPs to select candidate genes. To investigate the salt-tolerance mechanism, non-synonymous SNPs from Nampungchal were used. Among the DEGs between Nampungchal and Sodamchal under salt stress, the number of DEGs expressed in leaves is 533. Of these, 86 DEGs have non-synonymous-SNPs (Supplementary Table S1). On the other hand, there are 676 DEGs in the root, of which 86 DEGs have non-synonymous SNPs (Supplementary Table S2). These genes need to be focused on in terms of salt-tolerance because they are upregulated and change the structure of the protein, possibly contributing to the function of those proteins.

### 3.3.3. Selection Pressure of the Genes Calculated as the Ratio of $K_a/K_s$

The ratio of non-synonymous and synonymous substitution was evaluated to find out the direction of gene evolution under natural selection pressure. Genetically, the $K_a/K_s$ ratio has been used as an indicator of natural selection during the history of evolution. To effectively adapt to the changing environment, positive selection diversifies its genotypic variants for the advantage of being transferred to the next generation ($K_a/K_s > 1$). On the other hand, negative selection indicates genes that are unfavorable to adapting to the environment and that are removed ($K_a/K_s < 1$). Selection pressure was compared between the reference genome, BTx623, and the two Korean cultivars. The $K_a/K_s$ ratios of Nampungchal and Sodamchal were predicted by using our in-house python script. As a result, there were 3007 genes with a $K_a/K_s$ ratio higher than one in Nampungchal (Supplementary Table S3). On the other hand, in Sodamchal, 2261 genes had a $K_a/K_s$ ratio higher than one (Supplementary Table S4). The impact of the variants in genes with a $K_a/K_s$ ratio higher than one was compared in each cultivar. There was a total of 24,269 variants in the genes with a $K_a/K_s$ ratio greater than one in Nampungchal. There were 657 (2.71%) variants with a high impact, 2762 (11.38%) variants with a low impact, 7708 (31.76%) variants with a moderate impact and 13,142 (54.15%) variants with a modifier impact. On the other hand, in Sodamchal, the total number of variants of genes with a $K_a/K_s$ ratio greater than one was 19,519. Among them, there were 531 variants with a high impact (2.72%), 1862 variants with a low impact (7.62%), 5652 variants with a moderate impact (28.96%), and 11,474 variants with a modifier impact (58.78%). Moreover, to select candidate genes associated with salt stress, we looked into the selection pressure of the DEGs. The DEGs that were commonly upregulated in the leaves and roots were subjected to a selection-pressure analysis. Selection pressure was identified in 26 genes out of a total of 152 DEGs which were commonly upregulated in the leaves and roots. Among them, we found that a total of 15 and 9 genes were positively and negatively selected, respectively (Table 2).

**Table 2.** DEGs that were simultaneously upregulated in both leaves and roots in Nampungchal compared to Sodamchal under salt stress with Ks/Ka ratios and mutational impacts.

| Gene Symbol | Ka/Ks | High | Low | Moderate | Modifier | Description |
|---|---|---|---|---|---|---|
| LOC8058550 | 10 | 3 | 1 | 5 | 5 | Signal-recognition particle 54 kDa protein, chloroplastic |
| LOC8079195 | 4 | 0 | 1 | 2 | 4 | E3 ubiquitin-protein ligase EL5 |
| LOC8067380 | 4 | 0 | 0 | 2 | 1 | Uncharacterized LOC8067380 |
| LOC8060343 | 4 | 0 | 1 | 6 | 2 | Uncharacterized LOC8060343 |
| LOC110434949 | 4 | 0 | 2 | 2 | 0 | Uncharacterized LOC110434949 |
| LOC110429564 | 4 | 0 | 0 | 2 | 0 | Uncharacterized LOC110429564 |
| LOC8057624 | 2.8 | 1 | 2 | 7 | 19 | Protein HASTY 1 |
| LOC8059007 | 2 | 0 | 0 | 1 | 0 | Eukaryotic translation initiation factor 3 subunit G |

**Table 2.** *Cont.*

| Gene Symbol | Ka/Ks | High | Low | Moderate | Modifier | Description |
|---|---|---|---|---|---|---|
| LOC8066766 | 2 | 0 | 0 | 1 | 3 | Proactivator polypeptide-like 1 |
| LOC8055639 | 2 | 0 | 0 | 2 | 6 | Ethylene-responsive transcription factor ERF104 |
| LOC8066157 | 2 | 0 | 2 | 5 | 0 | Uncharacterized LOC8066157 |
| LOC110432337 | 2 | 0 | 0 | 1 | 1 | Uncharacterized LOC110432337 |
| LOC110432325 | 2 | 0 | 0 | 1 | 1 | Uncharacterized LOC110432325 |
| LOC8057276 | 1.2 | 0 | 2 | 3 | 1 | F-box/FBD/LRR-repeat protein At4g00160 |
| LOC8074840 | 1.171717 | 1 | 50 | 58 | 7 | Disease resistance protein RGA2 |
| LOC8068782 | 1 | 0 | 0 | 1 | 2 | Jasmonate O-methyltransferase |
| LOC8065187 | 1 | 1 | 1 | 1 | 0 | Chaperone protein dnaJ 13 |
| LOC8062760 | 0.8 | 0 | 2 | 2 | 6 | Uncharacterized protein C227.17c |
| LOC8085912 | 0.666667 | 0 | 1 | 1 | 6 | TIP41-like protein |
| LOC8076019 | 0.666667 | 0 | 1 | 1 | 0 | RING-H2 finger protein ATL8 |
| LOC8072250 | 0.666667 | 0 | 1 | 1 | 3 | Histone H2B.1 |
| LOC8066486 | 0.545455 | 0 | 6 | 3 | 12 | Uncharacterized LOC8066486 |
| LOC8072272 | 0.470588 | 1 | 9 | 5 | 9 | Momilactone A synthase |
| LOC8071656 | 0.444444 | 0 | 4 | 2 | 2 | SPX domain-containing protein 1 |
| LOC8054095 | 0.4 | 0 | 2 | 1 | 0 | Nicotianamine aminotransferase A |
| LOC8082038 | 0.285714 | 0 | 3 | 1 | 0 | Probable LRR receptor-like serine/threonine-protein kinase At4g36180 |
| LOC8078184 | 0 | 0 | 1 | 0 | 21 | Momilactone A synthase |
| LOC8074201 | 0 | 0 | 1 | 0 | 0 | Mediator of RNA polymerase II transcription subunit 8 |
| LOC8077219 | 0 | 0 | 1 | 0 | 8 | Chlorophyll a-b binding protein 1, chloroplastic |
| LOC110429806 | 0 | 0 | 1 | 0 | 4 | CBS domain-containing protein CBSX1, chloroplastic-like |
| LOC8066950 | 0 | 0 | 1 | 0 | 2 | Uncharacterized LOC8066950 |
| LOC8063059 | 0 | 0 | 2 | 0 | 15 | Uncharacterized LOC8063059 |
| LOC110433303 | 0 | 0 | 1 | 0 | 0 | Uncharacterized LOC110433303 |
| LOC110429560 | 0 | 1 | 1 | 0 | 0 | Uncharacterized LOC110429560 |

## 4. Discussion

Salt stress is a factor that has a negative effect on plant growth and reproduction. Because salt stress is a complex trait, finding out the genetic traits and genes associated with salt tolerance is key to understanding the mechanisms of the salt-stress response in plants. Sorghum is considered a drought-tolerant crop. Moreover, sorghum has been evaluated as a relatively salt-tolerant crop [33]. To understand the mechanism of salt stress, we conducted a study with two cultivars of salt-tolerant and salt-sensitive sorghum.

Anthocyanins are a large class of water-soluble pigments belonging to the flavonoid group and tend to be found in all tissues of the plant [34]. Anthocyanin accumulates differently in each tissue under various environmental stimulations. Anthocyanins are a good indicator of the stress response and have an important defensive role in the abiotic or biotic stress of plants from damage caused by reactive oxygen species in plants [35]. Salt stress tends to reduce the anthocyanin levels in salt-sensitive crops. On the other hand, the salt-tolerant genotype maintains the accumulation of anthocyanin, indicating a significant role in the plant's salt reaction [36]. Studies have reported that high anthocyanin contents improve salt tolerance in Arabidopsis [37] and wheat [38]. In this study, the anthocyanin contents of the roots showed a tendency to increase as salt stress progressed in both genotypes. However, it showed a different trend in the leaves, which had a relatively higher anthocyanin contents than that of the roots. In the leaves, the anthocyanin contents showed a tendency to gradually increase as the stress progressed for a long time. However, in Sodamchal, the anthocyanin contents in the leaves decreased as the salt stress progressed. Moreover, nine days after the salt-stress treatment, a significant difference between the two cultivars was found only in the leaves and not in the roots. This trend indicates that anthocyanins respond to stress more in the leaves than in the roots. This result is consistent

with previous studies in which anthocyanins increase the salt tolerance. The accumulation of anthocyanins in Nampungchal may affect the salt-tolerance by possibly acting as a scavenger to reactive oxygen species.

As salt-stress progresses, osmotic stress occurs due to the high sodium concentration. Osmotic stress causes dehydration in the plant and inhibits the growth and development of crops. An ionic imbalance caused by the accumulation of sodium in plants causes a deficiency of potassium and impairs the homeostasis of cells. During salt stress, plants synthesize proline, which is a cellular osmotic regulator that protects enzymes, proteins, membranes, and the structure of cells from external stress. Additionally, proline confers drought or salt tolerance through its ability to scavenge hydroxyl radicals [39]. In our study, we found that the proline contents of both sorghum cultivars increased in both the leaves and roots when subjected to salt stress. These results lead to similar conclusions as those reported in previous studies that the proline contents increase when plants such as rice [40], tomatoes [41], and potatoes [42] are subjected to salt stress. Overall, the salt-tolerant Nampungchal has lower proline contents than that of the salt-sensitive Sodamchal. In our case, the proline contents seemed to be associated with the degree of tolerance; however, it is not always the case. A previous study [43] researched the accumulation of proline in rice under salt stress. However, it was identified that the proline contents were higher in the salt-sensitive rice cultivar than in the salt-tolerant rice cultivar. Although proline is a response that occurs when a plant suffers salt-stress damage, it is not considered a criterion for salt tolerance. Another study [44] performed a comparative study of proline accumulation in two sorghum cultivars for salt tolerance and argued that proline accumulation is a response to salt stress but not an indicator of salt tolerance. Therefore, the proline contents in our study are somewhat associated with salt stress, indicating that it can be used as a phenotypic marker which identifies the existence of salt-stress in the early stage of growth.

In stressful situations, plants want to maintain the water potential and turgor in the plant and secure the water intake necessary for growth [45]. This mechanism enables plants to increase osmotic pressure through the synthesis of soil solutes or metabolic solutes [46]. Of the various organic osmoprotectants, sugars improve the total osmotic potential by about 50% in glycophytes under salt-stress conditions. An increase in soluble carbohydrate accumulation in plants has been reported in salt- or drought-stress conditions [47]. In particular, among carbohydrates, sugar (glucose, fructose, sucrose, and fructans) accumulates during salt stress, while having a role in osmotic protection, osmotic adjustment, carbon storage, and radical scavenging. It has been reported that the contents of these soluble sugars were significantly elevated in tomato leaves under salt-stress conditions [48]. Moreover, it has been reported that the accumulation of soluble sugars occurs differently depending on the salt tolerance of plants. According to a previous study, sugar was measured for five sunflower accessions, and it was reported that when the salt tolerance is higher, more sugar is accumulated [49]. In our study, the reducing sugar contents in the leaves of Nampungchal increased significantly after receiving salt stress (three days after treatment) and were maintained at a certain level over time. On the other hand, the reducing sugar contents in the leaves of the Sodamchal decreased nine days after the salt-stress treatment compared to before the salt-stress treatment. It was observed that the reducing sugar contents nine days after the salt-stress treatment in the leaves had a difference of more than two times between the two sorghum cultivars. It can be inferred that the reducing sugar acts as an osmoprotectant in the salt-tolerant Nampungchal. However, because the reducing sugar decreased over time in Sodamchal, it is probably related to salt sensitivity. On the other hand, the reducing sugar contents in the roots tended to remain constant even when subjected to salt stress in the Nampungchal cultivar. However, the salt-sensitive Sodamchal had increased the reducing sugar contents at the early stage of the salt-stress treatment (three days after the salt treatment), and it decreased as the salt stress continued (nine days after the salt treatment). These results identified that Nampungchal responds more stably to external stimuli than Sodamchal by increasing or maintaining the sugar contents in a stressful situation.

Chlorophyll is a green pigment found in algae or plant chloroplasts. Chlorophyll absorbs energy from light and is essential for the photosynthesis of plants. When plants are exposed to salt stress, reactive oxygen species increase, and damage inside the cells occurs [50]. Chlorophyll loss is measured by the production of malondialdehyde, which is known as an indicator of damage caused by reactive oxygen species [51]. Reactive oxygen species cause a decrease in the activity of chlorophyll. In response to salt stress, plants protect cells from oxidative damage by creating a non-enzymatic detoxification system (ascorbate, carotenoids, and flavonoids) [52]. Carotenoids, which have various functions, especially photosynthesis, possess a mechanism to defend against attacks by oxidative stress. The results of this study identified that there is an inverse relationship between the amount of chlorophyll and salt stress [53]. The reason that the chlorophyll contents are maintained or increased in the early days of salt stress is the probability that it is trying to protect the plant against the reactive oxygen species generated when salt stress occurs. However, as the damage caused by salt stress is prolonged, it seems that the cellular structure in the plant is destroyed due to the continuously accumulated reactive oxygen species, and thus, the chloroplasts are reduced. Overall, the salt-tolerant Nampungchal showed chlorophyll higher contents than that of the salt-sensitive Sodamchal, thus indicating that plant chlorophylls may be more damaged in the salt-sensitive accessions under salt-stress conditions.

Prior to the genomic analysis between the two sorghum cultivars, physiological studies were conducted on anthocyanin, chlorophyll, proline, and reducing sugar to determine the difference in physiological responses to salt stress. For each experiment, the trend was somewhat similar to the results of previous studies. Although physiological indicators can be considered as a response to salt stress, they are not criteria for determining salt tolerance. Therefore, for an advanced understanding of salt tolerance, we attempted to analyze the transcriptome data in many ways and to find candidate genes involved in salt tolerance.

To find candidate genes related to salt tolerance, we classified DEGs that were simultaneously upregulated in the leaves and roots of Nampungchal compared to Sodamchal after the salt-stress treatment. As a result, 152 DEGs were identified, of which 82 are genes whose exact descriptions are not yet known, and only 70 genes had their descriptions confirmed. Among the DEGs, 'LOC8078184' and 'LOC8072272' were described as 'momilactone A synthase'. Momilactone A is an allopathic phytoalexin that is generally known to have weed tolerance. A previously reported study identified that these proteins are strongly associated with drought and salt tolerance [54]. Another DEG, 'LOC8078573', is a gene encoding 'ricin B-like lectin R40C1'. Lectin proteins have a key role in responding to abiotic stresses in plants. It has been reported that rice overexpressing the lectin protein r40c1 has a higher level of drought tolerance than that of the wild type [55]. The 'LOC8085367' gene is a gene with the function of 'auxin-responsive protein IAA21'. Auxin has a significant role in the developmental responses of various cells in plants. Aux/IAA family members are involved in root development and shoot growth [56]. The 'LOC8068782' gene is described as jasmonate O-methyltransferase. Jasmonates have an essential role in mediating plant responses to several abiotic stresses, such as salinity stress. A previous study reported that exogenous jasmonates relieve salinity stress and elicit an antioxidant response in plants [57]. The 'LOC8068095' is a coding gene for 'osmotin-like protein'. Osmotin and osmotin-like protein are a kind of protein produced when plants adapt to environmental stress and are involved in osmotic regulation and stress response. In particular, osmotin is induced in a salinity situation, and salt tolerance is improved in osmotic transgenic plants [58]. These genes probably confer salt tolerance to Nampungchal. To identify the genomic mechanism of salt tolerance more accurately, additional research is needed on genes whose functions are unknown yet.

GO analysis confirms the function of the identified DEGs through relevant biological processes, molecular functions, and cellular components. We performed a GO analysis for a more diverse functional analysis of the previously identified salt-tolerance candidate genes. 'GO:0017111, nucleoside-triphosphatase activity', the term with the most genes linked to

it, belongs to the hydrolase family. This enzyme is involved in purine metabolism and thiamine metabolism. These enzymes are also called nucleoside triphosphate phosphohydrolases (NTPDases). Previous studies reported that the expression of NTPDases was upregulated in wheat under salt stress and had a vital role in the salt-stress response [59]. In addition, three genes were associated with 'GO:0003924, GTPase activity'. According to a recent study, it was reported that transgenic wheat overexpressed with MfARL1, a gene encoding GTPase, showed a high catalase activity, low chlorophyll reduction, and low accumulation of $H_2O_2$ and malondialdehyde under salt stress [60]. In KEGG analysis, pathways that can be related to photosynthesis, such as chlorophyll ab binding protein 2, chloroplastic, 'chlorophyll ab binding protein 1', and 'Photosynthesis antenna proteins', were identified. It is also involved in signaling and metabolic pathways, such as 'Biosynthesis of secondary metabolites' and 'Plant hormone signal transduction'. It also responds to abiotic stress through antioxidant metabolism, such as 'Flavonoid biosynthesis' and 'Arginine and proline metabolism'. Through these GO and KEGG pathway analyses, we confirmed that the candidate genes are directly or indirectly related to salt stress through the classification of pathways and molecular functions.

SNPs are single-nucleotide-sequence mutations that are generated by transitions or transversions. SNPs cause genetic polymorphisms and are distributed in intergenic regions, exons, introns, UTR, etc. SNPs in the coding region are divided into two types: synonymous and non-synonymous SNPs. Among them, non-synonymous SNPs affect protein function by fluctuating its three-dimensional structure [61]. The influence of gene expression or protein function can be caused by SNPs. Therefore, research on SNPs has exciting potential for genetic, breeding, and evolutionary studies. We used several bioinformatics tools to detect SNPs and built a pipeline for functional classification and analysis of the SNPs. We called SNPs from two cultivars, Nampungchal and Sodamchal, showing different responses to salt stress. The SNPs of the two sorghum cultivars were classified based on locations, impacts, and features. Particularly, we focused on non-synonymous SNPs that change the function of proteins. Among the DEGs upregulated in the salt-tolerant Nampungchal, genes with non-synonymous SNPs were selected. A total of 86 genes were found in the leaves and roots. Among them, 'LOC8086204' and 'LOC8073388' expressed in leaves are genes that encode a zinc finger protein, which mainly transforms and degrades stress-related proteins and is known to have tolerance to abiotic stress [62]. Three genes, 'LOC8063682', 'LOC8081774' in leaves, and 'LOC8069719' in roots, encode 'UDP-glycosyltransferase'. It has been reported that UDP-glycosyltransferase is induced by various abiotic stresses, including salinity stress, and that overexpression significantly increased plant tolerance to salinity [63]. Moreover, 'LOC8065088' in leaves, 'LOC8065187', 'LOC8065176', and 'LOC8065052' in roots function as a pentatricopeptide repeat-containing protein (PPR), and chloroplast PPR has been reported to regulate the response of plants to abiotic stresses [64]. The 'LOC8065176' gene expressed in the root encodes 'G-type lectin S-receptor-like serine/threonine-protein kinase' (GsSRK). GsSRK is a gene that responds to external signals and imparts tolerance to salt stress. A previous study reported that overexpression of GsSRK in Arabidopsis promoted germination and growth under salt stress [65]. These genes may have structural changes due to non-synonymous substitutions, possibly resulting in some changes from the original functions, in turn, contributing to the salt tolerance of Nampungchal. Although we may not be able to try to conduct functional studies for those genes because it is outside the scope of our research field, it is worthwhile for other researchers to try to perform molecular studies associated with salt-stress with the candidate genes we proposed in this current study.

In general, the $K_a/K_s$ ratio of orthologous genes indicates the evolutionary selection pressures of species [66]. Although the concept of this analysis is to see the patterns of long-time divergence, we tried to apply this concept for two closely related cultivars with the reference genome. This may offer some clues to infer artificial-selection pressure during their relatively short breeding history, providing some basis for the difference between Nampungchal and Sodamchal in terms of different responses to salt stress. In

our results, a total of 3007 and 2261 genes with a $K_a/K_s$ ratio higher than one were identified in Nampungchal and Sodamchal, respectively (Figure 9). This means that positively selected genes were found more frequently in Nampungchal than in Sodamchal. These two sorghums are cultivars, and the generation of these two cultivars occurred recently. Therefore, as stated, the difference in the $K_a/K_s$ ratio may be caused by the artificial selection undertaken by breeders rather than by natural selection. Thus, we should understand these $K_a/K_s$ ratios from a breeding point of view. Specific phenotypes of sorghum are selected by breeders during the breeding process. The difference in phenotype in the sorghum population is generally determined by genetic differences among sorghum individuals. Moreover, these genetic differences are likely caused by genetic variations such as non-synonymous SNPs that alter protein functions. We wanted to know how SNPs caused the genetic and phenotypic differences so that sorghum individuals were selected during the breeding process. To obtain the patterns of breeders' selection, we identified the $K_a/K_s$ ratios and selective pressures of genes differently expressed in the salt-tolerant Nampungchal and salt-sensitive Sodamchal. The $K_a/K_s$ ratios were identified in 26 out of 152 DEGs (which are simultaneously upregulated in both tissues of Nampungchal in contrast to Sodamchal under salt conditions). Among them, 15 genes have a $K_a/K_s$ ratio greater than one, while nine genes have a $K_a/K_s$ ratio less than one. (Table 2.). The genes selected by breeders are highly likely to be negatively selected ($K_a/K_s < 1$). In general, genetic-sequence changes are often more harmful than beneficial. Negative selection serves to maintain the stability of biological structures by removing harmful mutations. During the development of cultivars over several generations, the genetic structure of the plant is stabilized, and specific traits are fixed in the plant. In this selection process, negative selection, which has a role in stabilizing the genetic structure, will have a greater influence on the breeding process. Therefore, the selection pattern achieved by breeders is more likely to be associated with negatively selected genes. The gene 'LOC8076019' with a $K_a/K_s$ ratio of 0.667 encodes the protein RING-H2 finger protein ATL8. The RING finger protein has a vital role in plant adaptation to abiotic stress. Previous research reported that overexpression of the RING-H2 finger protein gene improves salt tolerance in rice [67]. The 'LOC8071656' gene with the SPX-domain-containing protein has a $K_a/K_s$ ratio of about 0.444. SPX gene families are involved in plant growth and development. They also have important roles in metabolism, such as nutritional stress, light signaling, and resistance to disease [68]. The 'LOC8082038' gene encodes the probable LRR receptor-like serine/threonine-protein kinase At4g36180. Leucine-rich repeat receptor-like kinases (LRR-RLKs) have a significant role in the stress response and plant growth and development. In a previous study, overexpression of LRR-RLKs in Arabidopsis increased the salt tolerance and oxidative-stress tolerance of plants. In other words, LRR-RLKs are reported to modulate stress signaling networks and confer abiotic-stress tolerance [69]. However, the genes causing the phenotypic difference between Nampungchal and Sodamchal are more likely to be positively selected genes than negatively selected genes. Nampungchal is a cultivar that targets lodging tolerance and high yielding ability, not salt tolerance. However, Nampungchal is more tolerant to salt stress than Sodamchal. In the breeding process, it is highly likely that the salt-tolerance-related genes were introduced accidentally by linkage disequilibrium (LD). According to the selective-sweep theory, positively selected mutations ($K_a/K_s > 1$) are fixed with a rapid increase in frequency. After that, LD is formed by removing and reducing variations around the beneficial mutations. Thus, positively selected genes in LD were possibly unconsciously selected during the breeding process, and these genes may be related to salt tolerance. Therefore, we checked the functions of positive selection genes, which are likely to be accidentally introduced by LD during the breeding process. The gene with the highest $K_a/K_s$ ratio of 10 is 'LOC8058550'. This gene encodes 'putative signal recognition particle 54 kDa protein, chloroplastic'. It is a precursor of chloroplast and involved in chloroplast development. Signal-recognition particles play an essential role in binding to the signal sequence of the nascent polypeptide and delivering the polypeptide to the appropriate cells of the chloroplast [70]. 'LOC8079195', with a

$K_a/K_s$ ratio of 4, has the function of 'E3 ubiquitin-protein ligase EL5'. E3 ubiquitin ligase modulates plant hormones and light signaling and is involved in regulatory pathways. E3 ubiquitin ligase can regulate plant development and abiotic-stress responses and has an important role in determining substrate specificity [71]. The 'LOC8059007' encodes 'eukaryotic translation initiation factor 3 subunit G' (elF3), with a $K_a/K_s$ ratio of 2. The elF3 is a large protein complex involved in most translation-initiation processes. Previously reported studies showed that OseIF3e identified in rice was highly expressed in actively growing organs, and OseIF3e-inhibited transgenic rice showed limited growth at the seedling and vegetative stages. Moreover, elF3 may play a vital role in plant growth and development by binding to other proteins [72]. The 'LOC8057276' gene has the function of 'F-box/FBD/LRR-repeat protein At4g00160', with a $K_a/K_s$ ratio of 1.2. The F-box family is one of the gene families involved in the entire life process of a plant. The F-box family is a subunit of the Skp1-Rbx1-Cul1-F-box protein (SCF) complex and has been identified in eukaryotes for decades. Some of the genes of the F-box family are involved in hormonal responses, light responses, and abiotic and biotic stress responses in plants [73]. These genes may have the potential to impart salt tolerance to plants. One thing to note is that, among the candidate genes, there are many genes whose exact function cannot be revealed yet. Therefore, further studies on these genes are needed and will be of significant help in elucidating the genetic mechanism of salt tolerance in sorghum.

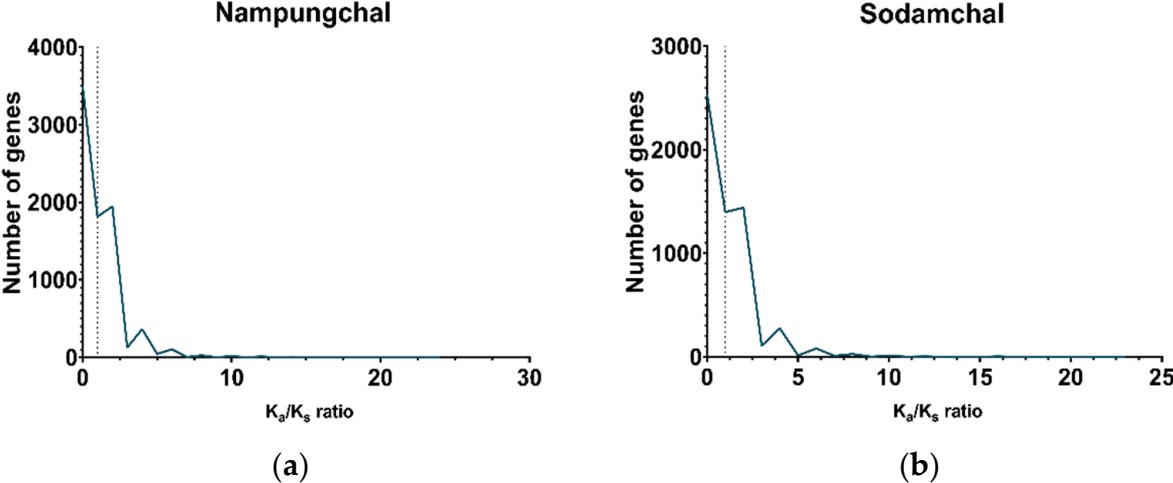

**Figure 9.** Distribution of the $K_a/K_s$ ratio of the genotype genes of the two sorghum cultivars: (**a**) Nampungchal and (**b**) Sodamchal. The *y*-axis represents the number of genes, and the *x*-axis represents the $K_a/K_s$ ratio of the genes. The dotted line represents a $K_a/K_s$ ratio of 1 which determines the selective pressure.

## 5. Conclusions

Salt stress inhibits plant growth and development worldwide. Sorghum is a genomic model crop of C4 plants and Saccharinae because of its small genome size. Sorghum is one of the important crops, with the fifth-most production among cereal crops in the world. Understanding the mechanisms of salt stress in sorghum will have immense value for its use as a biofuel and food resource in preparation for future climate change. However, despite the importance of sorghum, studies of the genetic mechanisms involved in the salt stress of sorghum are deficient. In our study, a comparative study was conducted with two sorghum cultivars having tolerance and sensitivity to salt stress. Prior to the comparative transcriptome study, anthocyanin, proline, chlorophyll, and reducing sugar contents were identified to compare the salt-stress response of sorghum. As a result, although these physiological contents do not determine salt tolerance, there were differences between the two cultivars. We performed QuantSeq on the two sorghum cultivars under salt stress and normal conditions. We identified 152 DEGs, which were commonly upregulated in the leaves

and roots between Nampungchal and Sodamchal. Out of a total of 152 DEGs, 15 were positively selected, and 9 were negatively selected. Taking into account the specificity of the cultivar, we reconstructed the evolutionary pressures of the genes from a breeding perspective. Twenty-four genes were not intentionally selected by breeders, but they caused a difference in salt resistance between the two sorghum cultivars. Negatively selected genes are genes that are fixed for a long time, while stabilizing the genetic structure of the plant. Actually, these genes were identified as LOC110429806 (chloroplastic-like), LOC8077219 (chlorophyll a/b binding protein), LOC8072250 (histone H2B.1), and LOC8071656 (SPX domain-containing protein), which are genes related to the maintenance of plant organs and homeostasis. In addition, genes that increase resistance to salt stress such as LOC8076019 (RING-H2 finger protein ATL8) and LOC8082038 (gene encodes the probable LRR receptor-like serine/threonine-protein kinase) were also identified. Moreover, positively selected genes are highly likely to be genes that directly make a difference in salt resistance between Nampungchal and Sodamchal. There is a possibility that genes related to salt tolerance are accidentally fixed in the breeding process through selective sweep after LD formation. In particular, LOC8058550 (putative signal recognition particle 54 kDa protein, chloroplastic), LOC8079195 (E3 ubiquitin-protein ligase EL5), LOC8059007 (eukaryotic translation initiation factor 3 subunit G), and LOC8057276 (F-box/FBD/LRR-repeat protein At4g00160) are associated with tolerance to abiotic stress. Through this in-depth analysis process, several candidate genes that can confer salt tolerance to the plants were presented. Furthermore, this comprehensive comparative analysis of the salt tolerance of sorghum will be a valuable resource in future salt-tolerant sorghum breeding programs and will be useful background information for genetic studies on C4 plants.

**Supplementary Materials:** The following supporting information can be downloaded at https://www.mdpi.com/article/10.3390/agronomy12102511/s1. Table S1: DEGs with non-synonymous SNPs in leaves. Table S2: DEGs with non-synonymous SNPs in roots. Table S3: List of Nampungchal genes, along with Ka/Ks ratio based on the sorghum reference genome (BTx623). Table S4: List of Sodamchal genes, along with the Ka/Ks ratio based on the sorghum reference genome (BTx623).

**Author Contributions:** Conceptualization, D.J. and C.K.; methodology, D.J.; software, D.J.; validation, D.J., Y.K. and C.K.; formal analysis, D.J.; investigation, D.J., S.L., and S.C.; resources, D.J., S.L. and S.C.; data curation, D.J.; writing—original draft preparation, D.J.; writing—review and editing, C.K.; visualization, D.J.; supervision, C.K.; project administration, C.K. All authors have read and agreed to the published version of the manuscript.

**Funding:** National Research Foundation of Korea (NRF, No. 2022R1A2C1004127).

**Institutional Review Board Statement:** Not applicable.

**Informed Consent Statement:** Not applicable.

**Data Availability Statement:** The datasets generated during and/or analyzed during the current study are available in the NCBI repository, https://www.ncbi.nlm.nih.gov/bioproject/PRJNA807064.

**Acknowledgments:** This work was supported by the National Research Foundation of Korea (NRF) grant funded by the Korean government (MSIT, No. 2022R1A2C1004127).

**Conflicts of Interest:** The authors declare no conflict of interest.

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
