# Peer review of "Identification of SNPs and Candidate Genes Associated with Salt Stress in Two Korean Sorghum Cultivars and Understanding Selection Pressures in the Breeding Process"

_agronomy, doi:10.3390/agronomy12102511_

Round 1

Reviewer 1 Report

General comment
In this research, the authors used QuantSeq to investigate how two Korean sorghum varieties reacted to salt stress to compare the physiological responses of salt-tolerant and -susceptible cultivars, pinpoint genes whose expression varies between the two types of plants, and also to dissect mutations that alter gene function. The study is very interesting and well designed, and the findings may give significant resources for functional genomic research and may aid in deciphering the Salicaceae's complicated evolutionary architecture. Actually, the findings of this work can be used as a starting point for further sorghum genetics and breeding studies, as well as to better understand the plant's overall response to salt stress. I have some minor comments which could be addressed to improve the manuscript and increase chance of publication.
• Introduction part lacks to include literature about effect of salininty using such traditional phenotypic tools, does not any one investigated effect of salinity on sorghum before?
• Introduction also did not include any info about adopting molecular work and modern tools to investigate salinity stress in general and on sorghum, in particular.
• Line 138: why you did not estimate chlorophyl content using green leaf via using devices.
• Line 184: please add reference to the web-based tool “PlantRegMap”, and also please provide the URL of this website like what you did with KEGG in line188.
• Line 220: what “SEM” stands for? It should be written full in letters not abbreviated, otherwise it was full written before.
• Did you also draw all figures in the manuscript using SPSS software?
• Line 323: Are you sure about this sentence “relative expression level of the physiological….”. Which expression you are talking about?
• Line 325: “different days ANOVA followed……...” no need to mention ANOVA here again because you can use TUKEY mean comparison test without conducting ANOVA test. You can write this sentence as follows: “The different letters indicate significant differences among different days using Tukey’s test at p < 0.05.”
• Line 355-356: “……of which 70 genes are accurately described for their protein function”, I think no need to mention that here, because you are going to state that later in the next subtitle “ 3.2.2. GO classification and KEGG pathway” by which you identified that they are protein function functional.
• Line 391: " yellow bar not yellow bars, it is only one term acting as a cellular component.
• In Figure 8e: Its “stop loss” not “stop lost”.
• In Figure 8e: please align all the figures items (a-g) to give a better vision.
• Line 515-518: this sentence should be replaced into discussion not in results.
• What is “G” letter in Table 2 in the description column indicate to? is it a typo? please check this table again.

Author Response

Thanks for your comments. Responses to comments are in the attachments. Please see the attachment

Reviewer 2 Report

The manuscript is dedicated to identification of genes involved in salt stress response in sorghum. The authors set adequate tasks of the study that were realized with appropriate methods. In general, the manuscript makes a good impression and the results presented can be used for further studies of salt stress response. However, some revisions could improve the quality of manuscript:

Lines 17-20: Please avoid to put the technical data in the Abstract.

Line 20:  “We also correlated”  - it is not correct. Please replace to “found correlation between..” or “analyzed correlation between…”

Line 189: section “2.11. Analysis of Quantitative RealTime-PCR (qRT-PCR)”. Please provide the primers’ data used for qRT-PCR in Supplementary Files.

Figure 4 (b). The numbers in the diagram are unreadable.

Line 396: section “Validation of differentially expressed genes by qRT-PCR”. Please provide the correlation between the qRT-PCR and QuantSeq data.

Figure 7. Please improve the image quality, the plots are unreadable.

The “Conclusions” section: This part should be rewrite. The authors listed what they have done and what they have obtained but didn't summarize the main scientific results of the study.

English editing is required.

Author Response

(The authors gave the same response as above.)
